# Natural Variation in *OASC* Gene for Mitochondrial O-Acetylserine Thiollyase Affects Sulfate Levels in Arabidopsis

**DOI:** 10.3390/plants12010035

**Published:** 2022-12-21

**Authors:** Anna Koprivova, Büsra Elkatmis, Silke C. Gerlich, Martin Trick, Andrea L. Harper, Ian Bancroft, Stanislav Kopriva

**Affiliations:** 1Institute for Plant Sciences, Cluster of Excellence on Plant Sciences, University of Cologne, Zülpicher Str. 47b, 50674 Cologne, Germany; 2John Innes Centre, Norwich Research Park, Colney, Norwich NR4 7UH, UK; 3Department of Biology, University of York, Heslington, York YO10 5DD, UK

**Keywords:** sulfur homeostasis, Arabidopsis, cysteine synthase c, sulfate content, single nucleotide polymorphisms

## Abstract

Sulfur plays a vital role in the primary and secondary metabolism of plants, and carries an important function in a large number of different compounds. Despite this importance, compared to other mineral nutrients, relatively little is known about sulfur sensing and signalling, as well as about the mechanisms controlling sulfur metabolism and homeostasis. Sulfur contents in plants vary largely not only among different species, but also among accessions of the same species. We previously used associative transcriptomics to identify several genes potentially controlling variation in sulfate content in the leaves of *Brassica napus*, including an *OASC* gene for mitochondrial O-acetylserine thiollyase (OAS-TL), an enzyme involved in cysteine synthesis. Here, we show that loss of *OASC* in *Arabidopsis thaliana* lowers not only sulfate, but also glutathione levels in the leaves. The reduced accumulation is caused by lower sulfate uptake and translocation to the shoots; however, the flux through the pathway is not affected. In addition, we identified a single nucleotide polymorphism in the *OASC* gene among *A. thaliana* accessions that is linked to variation in sulfate content. Both genetic and transgenic complementation confirmed that the exchange of arginine at position 81 for lysine in numerous accessions resulted in a less active OASC and a lower sulfate content in the leaves. The mitochondrial isoform of OAS-TL is, thus, after the ATPS1 isoform of sulfurylase and the APR2 form of APS reductase 2, the next metabolic enzyme with a role in regulation of sulfate content in Arabidopsis.

## 1. Introduction

The essential nutrient sulfur possesses versatile functions in the plant by forming the amino acids cysteine (Cys) and methionine (Met), different coenzymes, and prosthetic groups, and participating in the structure of defense molecules, such as glucosinolates (GLS) or camalexin [1]. Cysteine acts as a key molecule in the participation of sulfur in metabolism and connects sulfur, nitrogen, and carbon assimilation [2,3]. The bioavailable form of sulfur for plants is sulfate [1,4]. Sulfate taken up from the soil solution by roots is translocated to the leaves by the nutrient flow of the xylem, facilitated by sulfate transporters [5]. For cysteine synthesis, sulfate must be reduced. The sulfate transported into the cells is first activated to APS by the ATP sulfurylase (ATPS) enzyme [6,7]. Sulfide is formed by the reduction of APS by the subsequent action of APS reductase (APR) and sulfide reductase (SIR) [8]. The carbon backbone of cysteine comes from O-acetylserine (OAS), which is formed after the reaction of serine with Coenzyme A, catalyzed by serine-acetyl transferase (SAT) [5,9]. The O-acetylserine(thiol)lyase (OAS-TL) replaces the acetyl group in the OAS with sulfide [9]. SAT and OAS-TL enzymes form the cysteine synthase complex (CSC) [10]. Various isoforms of SAT and OAS-TL enzymes are found in different subcellular structures of the cell [5,11,12]. For example, the well-studied OAS-TL isoforms OAS-TL A, OAS-TL B, and OAS-TL C are localized in the cytoplasm, plastids, and mitochondria, respectively [6,10,13]. Similarly, SAT also is present in these compartments; therefore, all three major subcellular structures possess cysteine synthase complexes and have the potential to synthesize cysteine [10]. However, the contribution of the three compartments to total cysteine synthesis in the cell is not the same, and the main site for cysteine synthesis is the cytosol [14]. Interestingly, this requires a precise coordination of the metabolite fluxes in the cell, because sulfate reduction is specific to plastids, and the majority of OAS is produced in the mitochondria [3].

While there is a good understanding of the functions of genes involved in sulfur metabolism, identifying genes regulating sulfur homeostasis, sensing, and signalling still lags. However, given the importance of sulfur for crop plants, finding such regulatory genes is critical to underpin the development of crop varieties with improved sulfur nutrition. Using natural variation in sulfur content is a promising approach to reveal genes that contribute to controlling sulfur homeostasis. Indeed, in experiments with the model plant *Arabidopsis thaliana,* genetic variations in genes encoding the APR2 isoform of APS reductase and the ATPS1 isoform of ATPS were found to underlie differences in sulfate and/or total sulfur content [7,15,16]. In addition, varieties of the crop plant *Brassica napus* were investigated for variation in nutrient content using associative transcriptomics (AT) [17]. The analysis suggested that the gene encoding OAS-TL C (*OASC*) controlled sulfate levels in *B. napus* [17]. This was confirmed by showing that the T-DNA mutant of the *OASC* gene in Arabidopsis resulted in significantly reduced sulfate accumulation in the leaves [17]. Here, we show that the natural variation in the *OASC* gene is also linked to variation in sulfate content in Arabidopsis accessions. We identified a non-synonymous single nucleotide polymorphism responsible for this variation, as evidenced by both genetic and transgenic complementations. However, the mechanism by which *OASC* affects sulfate levels still needs to be elucidated.

## 2. Results

### 2.1. Characterization of the oasC Mutant

We previously showed that disruption of the *OASC* gene in Arabidopsis leads to reduced accumulation of sulfate [17]. Further metabolite analyses were, therefore, performed in the roots and leaves of Col-0 and *oasC* to understand how the disruption of the *OASC* gene affects the content of sulfur-containing biomolecules. As observed previously with greenhouse grown plants and also in seedlings grown on agarose plates, the loss of *OASC* resulted in significantly less sulfate accumulation in the leaf than in the wild-type Col-0 (Figure 1A). However, the absence of functional *OASC* in the root had no effect on sulfate accumulation (Figure 1E). Loss of *OASC* did not affect Cys content in the leaves; however, there was a decrease in glutathione concentration (Figure 1B,C). As for sulfate, the decrease in GSH content was not observed in the root (Figure 1F). These results strongly indicated that the sulfate assimilation in *oasC* is affected, specifically in leaves. However, the GLS levels were not affected (Figure 1D); hence, the alterations might be limited to primary sulfate assimilation.

To determine the cause of the lower accumulation of sulfate and GSH in the mutant, sulfate uptake and flux through the sulfate assimilation pathway were determined. Indeed, *oasC* showed a dramatic reduction in sulfate uptake of almost 30% compared to Col-0 (Figure 2A). The reduction of uptake in *oasC* also directly affected the translocation of sulfate to leaves, which was reduced to a similar extent (Figure 2B). Measurement of flux through the pathway, determined as incorporation of ^35^S into thiols and proteins, however, revealed no alteration in the *oasC* mutant compared to the wild type (Figure 2C,D). Therefore, the reduction in sulfate and GSH contents seem to result from the considerable drop in sulfate uptake. These results also showed that the loss of the *oasC* gene affects sulfate uptake.

Expression analysis was performed using RNA isolated from roots and leaves to test whether genes for sulfur metabolism were affected by the loss of the *OASC* (Figure 3). The transcript levels of the three isoforms of APR, the key enzyme in the reduction of sulfate, were not significantly affected by the disruption of the *OASC* gene, neither in the shoot nor in the root (Figure 3). The expression of two markers of sulfur deficiency, *SDI1*, which is involved in GLS regulation, and *GGTC2;1*, which contributes to GSH degradation, were also not different in *oasC* compared to Col-0 [4,18]. In addition, the expression levels of the major isoforms of OAS-TL were compared. As expected, no expression of the *OAS-TL C* isoform was detected in the shoots or roots of the *oasC* mutant (Figure 3). The transcript level of the cytosolic *OAS-TL A* was slightly, but significantly, elevated in the shoot, but not in the root. In contrast, no changes in the transcript levels of the plastidic *OAS-TL B* or the mitochondrial *OAS-TL C1* isoforms were observed in *oasC* compared to Col-0 (Figure 3). Thus, in general, the loss of the *oasC* gene had almost no effect on the expression of genes for sulfate assimilation.

Next, we wanted to see if the reduction in sulfate levels in the *oasC* mutant is specific to this OAS-TL isoform, or whether loss of other isoforms might have the same effect. We, therefore, measured sulfate concentrations in the shoot of mutants in six OAS-TL isoforms in Arabidopsis and the double mutants of the main A, B, and C forms, and, in addition, in the *sat2;1* mutant in the mitochondrial form of SAT. Among all the mutants, only *oasC* and *oasC1* caused a significant decrease in sulfate accumulation (Figure 4). In contrast, in *oasA, oasAB, oasBC* and *oasD2*, sulfate accumulation in the shoot was slightly increased compared to Col-0. Thus, only changes in mitochondrial OAS-TL have a negative effect on sulfate accumulation. In roots, only *oasB* showed a small, but significant, increase in sulfate accumulation compared to Col-0. The mitochondrial *serat2.1* knockout mutant had no significant effect on sulfate levels compared to Col-0, indicating that the changes in sulfate are not caused by the cysteine synthase complex, but are specific to OAS-TL. To test whether the effect of loss of *OASC* on sulfate levels is dependent on external sulfate supply, various sulfate concentrations were supplied to Col-0 and *oasC*. Interestingly, the sulfate accumulation in *oasC* and Col-0 was affected differently by low and high sulfur supply. As in previous experiments, sulfate accumulation in the *oasC* was reduced compared to Col-0 when 750 µM sulfate was supplied. However, when 15 µM sulfate was supplied, the loss of *OASC* resulted in significantly higher sulfate accumulation than in Col-0 (Figure 5). Thus, *OASC* seems to have different regulatory roles depending on the sulfate status of the plant.

### 2.2. Amino Acid Variation in OASC Responsible for Variation in Sulfate Level

The *OASC* gene A_JCVI_8073 was identified in an associative transcriptomics with *B. napus* as linked to variation in sulfate content [17]. To find out the responsible genetic variation, the SNP markers within the *OASC* gene in 84 *B. napus* varieties used for the AT analysis were examined and connected to the sulfate content measured in Koprivova et al. [17] (Appendix A). Six haplotypes were found to be associated with a difference in sulfate content. However, only one of them, JCVI_8073:377, was a non-synonymous SNP, changing a Q_335_ (the position corresponding to AtOASC) to R, and, therefore, likely to be the causative SNP (Appendix A). While other SNPs were linked to the JCVI_8073:377, they represented either synonymous SNPs or were located in the 3’non-translated region on the transcript. Thus, it seems that a Q_335_R amino acid change in the *OASC* gene might be responsible for at least part of the variation of sulfate levels in *B. napus* varieties.

We then interrogated the sequences of Arabidopsis accessions from the 1001 genome project [19] for variation in the *OASC* gene. Interestingly, another non-synonymous SNP, leading to a K_81_R amino acid alteration, was found in Ha-0 and other accessions. We collected 34 Arabidopsis accessions differing in the K_81_R haplotype and measured their foliar sulfate level (Figure 6A). While there was clearly a variation among the different genotypes, on average, the haplotypes with the Ha-0 allele had significantly lower sulfate content than haplotypes with the Col-0 allele (Figure 6B). The 18% reduction in sulfate levels between the two haplotypes was very similar to the 16% reduction in sulfate level in *oasC* compared to Col-0 (Figure 1A). The K_81_R variation may, thus, be associated with the low sulfate content in *oasC*.

To confirm that the K_81_R causes the variation in sulfate between the two haplotypes, we employed genetic complementation. *oasC* and Col-0 were reciprocally crossed with Ha-0 and Col-0, and the F1 plants were examined. The sulfate levels of plants obtained by crossing *oasC* with Col-0 accumulated significantly more sulfate than plants obtained by crossing of *oasC* and Ha-0 (Figure 6C). Thus, clearly, the Ha-0 allele was not able to complement the loss of the *OASC* gene in the *oasC* mutant. To confirm that it is indeed the K_81_R variation that underlies the different functionality of the Col-0 and Ha-0 *OASC* genes, we performed a transgenic complementation of the *oasC* mutant. To avoid confounding the effects of the other variations between the accessions, we engineered the K_81_R SNP in the *OASC* gene from Col-0. Complementing *oasC* with the Col-0 allele of *OASC* (K_81_) significantly increased the sulfate level compared to *oasC* reaching those in the Col-0 wild type (Figure 7). In contrast, the Ha-0 allele with an R_81_ did not significantly increase sulfate compared to *oasC*. Thus, the K_81_R amino acid variation reduces the function of the *OASC* gene, resulting in a decrease in the sulfate level. However, since the K_81_ is part of the predicted organellar targeting peptide, the mechanism by which this variation affects *OASC* still needs to be elucidated, as well as the mechanism by which sulfate content is decreased in plants with the less active *OASC* alleles.

## 3. Discussion

Sulfur has an indispensable place in plant development in general and in the productivity of crops in particular [20,21]. For example, the sulfur-containing amino acid methionine is essential for animal and human nutrition, sulfur-containing secondary metabolites often promote health, and sulfur deficiency leads to a higher susceptibility to diseases, as well as to the accumulation of acrylamide in baked products from such crops [1,8,20,21,22]. *B. napus* is used in many different sectors, from livestock to cosmetics, and needs higher amounts of sulfur than other crop plants [23,24]. Therefore, *B. napus* with insufficient sulfur content shows a yield penalty. This is potentially a serious problem, because in the not-too-distant future, it is expected that abiotic and biotic stress factors, especially global warming, will decrease the availability of various nutrients, such as sulfur in the soil [25]. Therefore, there is a need to elucidate the regulation of genes involved in sulfur homeostasis for the production of *B. napus* (and other) crop plants with improved sulfur content. Indeed, several studies assessed the variation of ionome in *B. napus*, either in different varieties [26] or in response to perturbations in nutrient supply [27,28]. Another approach to learning about the control of nutrient homeostasis in *B. napus* was an associative transcriptomics study that identified several candidate genes controlling variation in nitrate, phosphate, and sulfate contents [17]. One of the candidate genes was the *OASC* for mitochondrial OAS-TL, which was the object of this study aimed at understanding how variation in *OASC* affects sulfate levels.

In Arabidopsis, OAS-TL is encoded by a multigene family with isoforms in different compartments [11,29]. Interestingly, the impact of loss-of-function mutations on growth is not consistent. While Watanabe et al. [11] did not observe any visible symptoms, Heeg et al. [29] described a reduction in growth of *oasB* and *oasC* mutants. Both studies agree on the loss of cytosolic OAS-TL A having the greatest impact on the total enzyme activity and concentration of thiols [11,29]. Watanabe et al. [11], but not Heeg et al. [29], showed a reduction in GSH content in *oasC* mutants, as measured also in our experiments (Figure 1). In contrast, the reduction in sulfate levels in *oasC* observed already in Koprivova et al. [17] and confirmed here (Figure 1) has not been observed before. Corresponding to the literature, however, *oasA* showed lower sulfate accumulation in this study (Figure 4) and in Heeg et al. [29]. Importantly, however, the reduced sulfate content in *oasC* was observed both in seedlings grown on nutrient solution and in plants grown in a greenhouse, pointing to a robustness of the phenotype.

The significantly decreased sulfate accumulation in *oasC* could be explained by reduced sulfate uptake or by increased sulfate utilisation. Given that GSH content was also reduced in *oasC* mutants (Figure 1), the latter explanation is most probably not correct. Indeed, feeding radioactively marked sulfate revealed not only reduced sulfate uptake and translocation to leaves, but also no changes in the rate of sulfate reduction (Figure 2). Therefore, the loss of *OASC* affects sulfate uptake, which in turn results in lower sulfate accumulation. This regulation is specific for OASC and is not compensated by other major OAS-TL isoforms (Figure 3). The reduction in sulfate, however, does not seem to be substantial enough to trigger a sulfate deficiency response, as the markers for sulfate deficiency, upregulated in other mutants with lower sulfate content [30], were not affected in this study (Figure 3). In addition, the effect of loss of OASC on sulfate is reversed during sulfate deficiency, when the mutant contains higher sulfate levels than WT. Although the interplay of the deficiency and the mutation still needs to be elucidated, this points to a better sulfur-use efficiency of the mutants. However, before it could be considered a way to improve the efficiency in crops the effects of the loss of OASC on fitness needs to be carefully assessed. Interestingly, among the SAT isoforms, it is also the mitochondrial SERAT2;2 that has the greatest impact on plant sulfur metabolism and growth [31]. Cysteine synthesis is a critical step in the incorporation of sulfate into sulfur-containing molecules [1,2,6]. The exact mechanism how the mitochondrial cysteine synthase regulates plant sulfur homeostasis, however, still needs to be elucidated.

Apart from the mechanisms explaining the effect of *oasC* mutation on sulfate content, the question on the exact nature of the natural variation in this gene leading to the variation in sulfate was addressed. The haplotype analysis on *B. napus* identified only one non-synonymous SNP, correlating with the sulfate content (Appendix A). Whether this sequence alteration indeed changes the enzymatic properties of OASC, however, needs to be determined. We instead turned to Arabidopsis and identified another SNP linked to variation in sulfate content. Accessions with a K at position 81 of the OASC sequence (Col-0-like) accumulated sulfate to higher levels than those with R (Ha-0-like) at this position (Figure 6). This result was confirmed by genetic and transgenic complementation analyses (Figure 6 and Figure 7), unequivocally showing that the K_81_R alteration affects the function of OASC. However, the K_81_ is located in the transit peptide, and, therefore, it is not clear what functional impact this variation may have. Possibly, it may affect the transport of the OASC precursor protein to the mitochondria. However, this hypothesis still needs to be tested.

In conclusion, we revealed that the loss of the *OASC* gene for the mitochondrial isoform of OAS-TL results in a significant decrease in sulfate content, unlike other OAS-TL mutations. The reduction in sulfate content is caused by a direct drop in the rate of sulfate uptake and translocation to the leaves. We also identified a K_81_R variation in *OASC* among Arabidopsis accessions that affects sulfate accumulation. Our findings, thus, pave the way towards a better understanding of sulfur homeostasis and the role of the *OASC* gene in sulfur metabolism.

## 4. Materials and Methods

### 4.1. Plant Materials and Growth Conditions

*Arabidopsis thaliana* L. ecotype Columbia-0 (Col-0) and T-DNA insertion line SALK_000860 that disrupts *OASC* (At3g59760; Appendix A), as well as additional *oastl* and *serat2;1* T-DNA insertion lines, were obtained from the Nottingham Arabidopsis Stock Centre (NASC) (Appendix A). Seeds were surface-sterilized with chlorine gas for 4 h. Under sterile conditions, seeds were placed on modified Long Ashton Medium agarose plates with 0.75 mM MgSO_4_ and stratified for 3 days at 4 °C in the dark. For sulfate deficiency experiments, the medium contained 15 µM or 45 µM sulfate, and the Mg^2+^ concentration was kept constant by adding 735 µM or 705 µM MgCl_2_, respectively. Afterward, the plates were incubated in Sanyo growth cabinets at 22 °C in long-day conditions, with a 16/8 h light cycle and 100 µmol photons m^−2^ s^−1^. After 18 days, shoot and root samples were collected and immediately frozen in liquid nitrogen. Each biological replicate was collected from seedlings grown on different plates. Results were obtained from at least two independent experiments.

The SNP molecular markers in *OASC* gene sequences of 84 varieties of *B. napus* were obtained from Harper et al. [32]. Arabidopsis accessions with Col-0 or Ha-0 alleles of the *OASC* gene were selected from the 1001 genome project and obtained from NASC. F1 plants were obtained after crossing Col-0 and *oasC* with Ha-0 and Col-0. For transgenic complementation, *OASC* gene sequence including 1500 bp upstream promotor sequence was amplified from *A. thaliana* Col-0 genomic DNA by PCR, cloned into pENTR-TOPO, and completely sequenced to exclude PCR artefacts. The A_242_ nucleotide was mutated to G by site-directed mutagenesis to create the K_81_R Ha-0 like allele (AAG into AGG). Both constructs were transferred into pGWB3 vector by LR clonase reaction. Transgenic plants were selected by hygromycin, and three independent transgenic lines were further analysed. For sulfate measurements, the plants were grown in the soil in the greenhouse for four weeks.

### 4.2. Measurements of Sulfur-Containing Metabolites

Sulfate levels were measured in root and leaf material by ion chromatography, exactly as described in Dietzen et al. [33]. The plant material was homogenized in 1 mL deionized H_2_O, shaken for 1 h at 4 °C, and then heated at 95 °C for 15 min. Inorganic anions were measured with the Dionex ICS-1100 chromatography system and separated on a Dionex IonPac AS22 RFIC 4 × 250 mm analytic column (Thermo Scientific, Darmstadt, Germany), using 4.5 mM Na_2_CO_3_/1.4 mM NaHCO_3_ as running buffer [33]. Cysteine and GSH concentrations were measured after conjugation to monobromobimane, using HPLC as described in Dietzen et al. [33]. The thiols were extracted from ca. 20 mg plant material in the 10-fold volume of 0.1 M HCl. Then, 25 μL extract were incubated with 25 µL of 0.1 M NaOH and 1 µL of 100 mM dithiothreitol (DTT) for 15 min at 37 °C in the dark. Subsequently, 10 μL 1 M Tris/HCl pH 8.0, 35 μL water, and 5 μL 100 mM monobromobimane (Thiolyte^®^ MB, Calbiochem) were added for 15 min at 37 °C in the dark. Next, 100 μL of 9% acetic acid was used to stabilize the bimane conjugates, which were separated and measured via high-performance liquid chromatography (HPLC; Spherisorb™ ODS2, 250 × 4.6 mm, 5 µm, Waters, Eschborn, Germany) and detected fluorimetrically (excitation: 390 nm, emission:480 nm), using a linear gradient of methanol in 0.25% acetic acid pH 3.9 [33]. GLS levels in leaves were determined as described in Huseby et al. [34]. The leaf material was extracted with 2 times 250 µL hot 70% MeOH, with the addition of 10 µL sinigrin as internal standard, and incubated at 70 °C for 45 min. After centrifugation, the extracts were loaded on DEAE Sephadex A-25 columns and washed twice with 0.5 mL dH_2_O and twice with 0.5 mL 0.02 M sodium acetate buffer. Then, 75 µL sulfatase solution was added on the surface of the column, and the columns were incubated overnight at room temperature. The produced desulfo-glucosinolates were eluted twice with water and were measured via HPLC (Spherisorb™ ODS2, 250 × 4.6 mm, 5 µm, Waters, Eschborn, Germany) by UV absorption at 229 nm, using a gradient of acetonitrile in water (5–30% in 8 min, 30–50% in 7 min), identified by retention time of the peaks, and quantified with the help of the internal standard sinigrin [34].

### 4.3. Analysis of Sulfate Uptake and Flux

Sulfate uptake and flux through the assimilation pathway were measured in 18-day-old seedlings grown on full sulfur supply using [^35^S] sulfate. The seedlings were incubated in 24-well plates in 1 mL nutrient solution containing 0.2 mM sulfate, supplemented with 12 µCi [^35^S] sulfuric acid, for 3 h in the light. Shoots and roots were extracted separately in a 10-fold volume of 0.1 M HCl. Ten microliters of extract were used to determine sulfate uptake, and 50 µL aliquots of the extracts were collected for quantification of [^35^S] incorporation into thiols, proteins, and glucosinolates, exactly as in [35].

### 4.4. RNA Extraction and Quantitative PCR Analysis

Total RNA was isolated by standard phenol/chloroform/isoamyl alcohol extraction and LiCl precipitation. First, strand cDNA was synthesized from 800 ng of total RNA, using QuantiTect Reverse transcription Kit (Qiagen, Hilden, Germany). Quantitative real time RT-PCT (qPCR) was carried out, using gene specific primers (Appendix A) and the fluorescent dye SYBR Green (Promega, Walldorf, Germany), as described in Koprivova et al. [36]. The expression level of genes was normalized, according to the *TIP41* (AT4G34270) gene. The qPCR reactions were performed in duplicate for each of the 4 independent samples.

## Figures and Tables

**Figure 1 plants-12-00035-f001:**
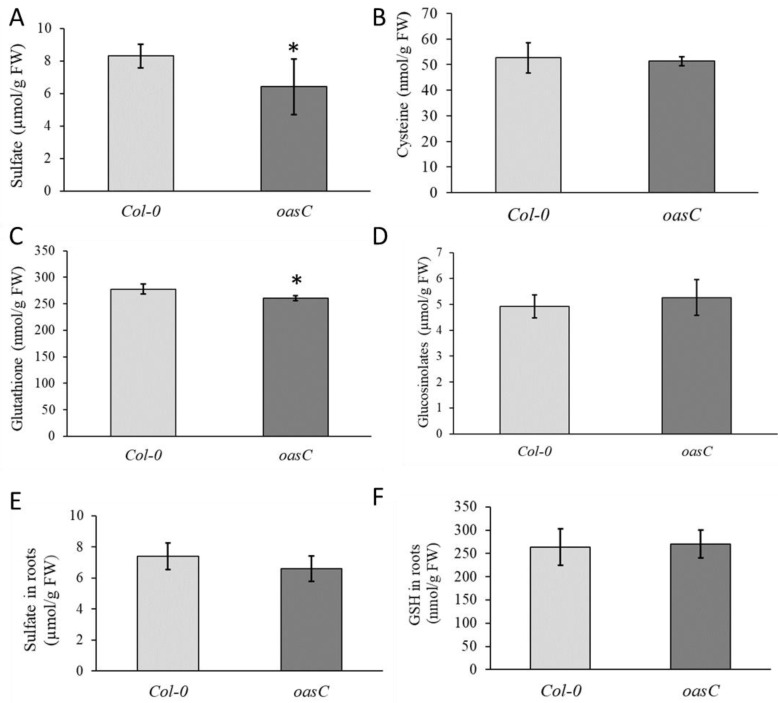
Disruption of *OASC* affects sulfur-containing metabolites. Col-0 (light grey) and *oasC* (dark grey) plants were grown for 2.5 weeks on MS-agarose plates. The content in the leaves of (**A**) sulfate, (**B**) cysteine, (**C**) glutathione, and (**D**) glucosinolates, as well as of (**E**) sulfate and (**F**) glutathione in roots was measured. Data are presented as means ± S.D. from four biological replicates. Asterisks mark values significantly different from the wild-type Col-0 at *p* < 0.05 (Student’s *t*-test).

**Figure 2 plants-12-00035-f002:**
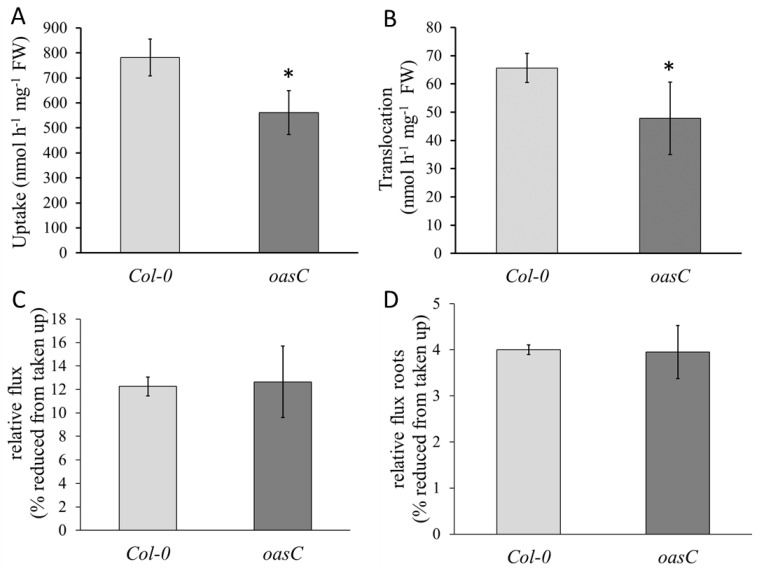
Sulfate uptake and translocation decreased in *oasC* mutant. Col-0 (light grey) and *oasC* (dark grey) seedlings grown on MS-agarose plates for 2.5 weeks were fed with the solution containing ^35^SO_4_^2−^ for 4 h. (**A**) Sulfate uptake and (**B**) translocation to shoots was determined by scintillation counting. Relative flux was measured as incorporation of [^35^S] in (**C**) thiols and proteins and (**D**) from the [^35^S] sulfate taken up. Data are presented as means ± S.D. from four biological replicates. Asterisks mark values significantly different from the wild-type Col-0 at *p* < 0.05 (Student’s *t*-test).

**Figure 3 plants-12-00035-f003:**
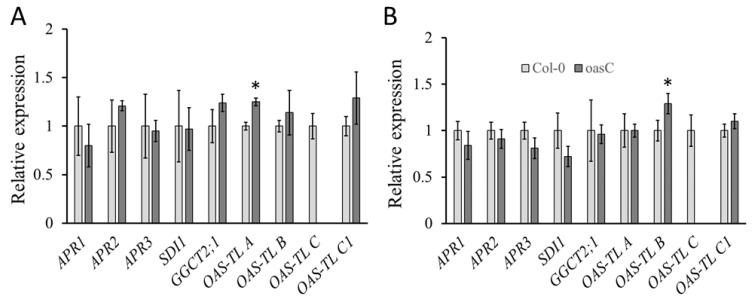
Disruption of *OASC* has no effect on expression of genes involved in sulfur metabolism. Col-0 (light grey) and *oasC* (dark grey) were grown for 2.5 weeks on MS-agarose plates. RNA was extracted from (**A**) shoots and (**B**) roots and relative transcript levels of nine genes involved in sulfur metabolism were analyzed by qPCR. The *TIP4* gene was used as an internal control to normalize expression levels. Data are presented as means ± S.D. from four biological replicates analyzed in duplicates. Asterisks mark values significantly different from the wild-type Col-0 at *p* < 0.05 (Student’s *t*-test).

**Figure 4 plants-12-00035-f004:**
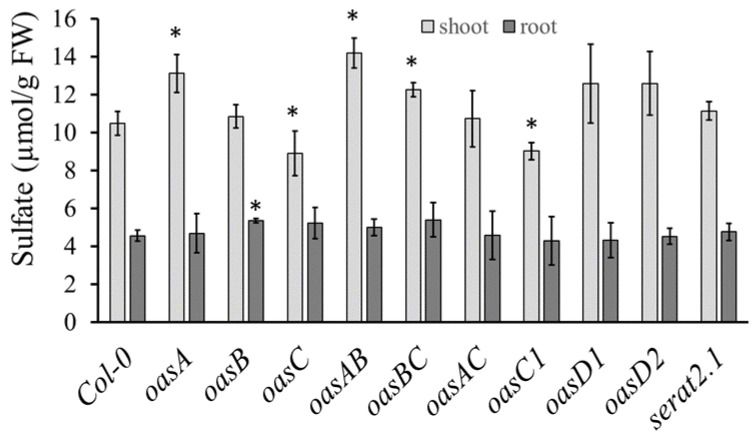
Sulfate accumulation is affected by the mutation of genes for OAS-TL isoforms. Col-0, *oasA, oasB, oasC, oasAB, oasAC, oasBC, oasC1, oasD1, oasD2* and *serat2.1* mutant plants were grown on MS-agarose plates for 2.5 weeks. The sulfate concentration of root and shoot was determined. Data are presented as means ± S.D. from four biological replicates. Asterisks mark values significantly different from the wild-type Col-0 at *p* < 0.05 (Student’s *t*-test).

**Figure 5 plants-12-00035-f005:**
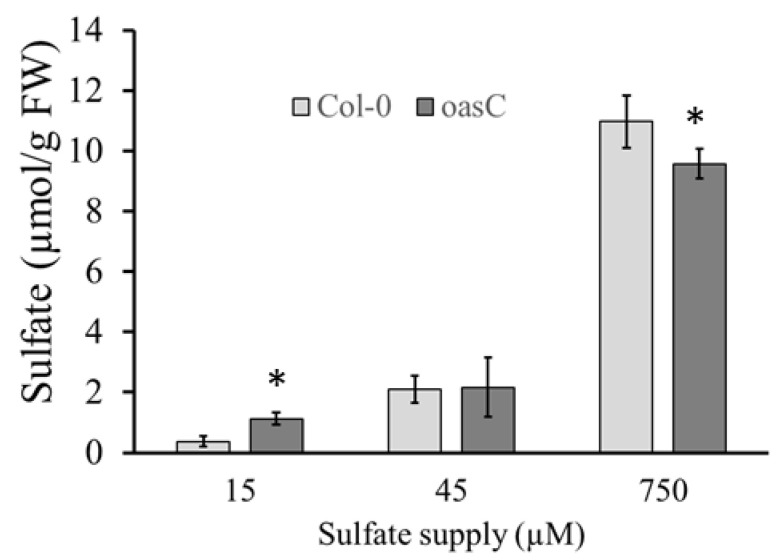
Exogenous sulfate supply affects sulfate levels in *oasC* mutant. Col-0 and *oasC* were grown on MS-agarose plates containing 15 µM, 45 µM and 750 µM sulfate. Sulfate levels of the leaves were measured. Data are presented as means ± S.D. from four biological replicates. Asterisks mark values significantly different from the wild-type Col-0 at *p* < 0.05 (Student’s *t*-test).

**Figure 6 plants-12-00035-f006:**
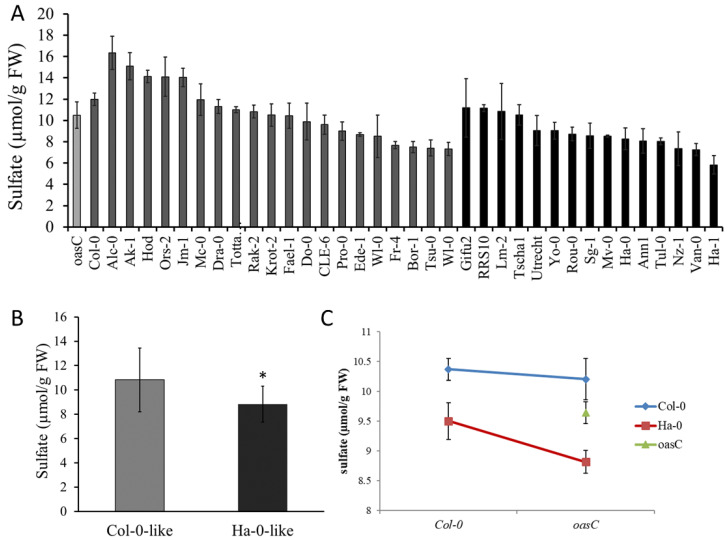
Sulfate content is affected by the Ha-0 allele of *OASC*. Plants were grown in soil in the greenhouse for four weeks. Shoots were harvested, and sulfate levels were measured. (**A**) Sulfate levels in different Arabidopsis accessions. Accessions with Col-0-like allele of *OASC* are dark grey, those with Ha-0-like allele are black. (**B**) Mean sulfate levels from Arabidopsis accessions with the two alleles of *OasC* from (**A**). Asterisks mark significantly different values at *p* < 0.05 (Student’s *t*-test). (**C**) Genetic complementation. Col-0 and *oasC* plants were crossed with Ha-0 and Col-0, F1 plants were grown in greenhouse for four weeks and sulfate was measured in the leaves. Data are presented as means ± S.D. from four biological replicates.

**Figure 7 plants-12-00035-f007:**
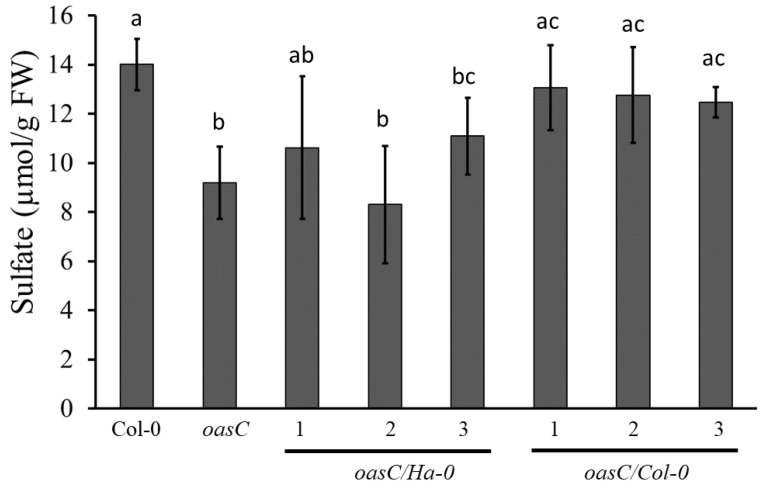
Transgenic complementation. Homozygous *oasC* mutants complemented either with Ha-0 or Col-0 allele of *OASC* were grown in soil in the greenhouse for four weeks. Sulfate levels in leaves were measured. Three lines were used for each construct. Data are presented as means ± S.D. from four biological replicates. Different letters denote significantly different values at *p* < 0.05 (Student’s *t*-test).

## Data Availability

All data is contained within the article or Appendix A. Additional material can be obtained on request to skopriva@uni-koeln.de.

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
