# Peer review of "Natural Variation in OASC Gene for Mitochondrial O-Acetylserine Thiollyase Affects Sulfate Levels in Arabidopsis"

_plants, 2022, doi:10.3390/plants12010035_

Round 1
Reviewer 1 Report
In this study, authors studied the function of OASC gene involved in sulfate uptake and translocation in Arabidopsis using the mutant. Additionally, they found a SNP in OASC gene among different accessions which is linked to the variation of sulfate content.
1. In the “Materials and methods” section, the detailed primer sequences for the gene amplification and gene expression analysis should be provided. The Supplementary Table S1 did not provide the primers.
2. The T-DNA insertion site of the mutants should be showed, as well as the gene expression level in mutants and wt plants.
3. In Fig.5, why the sulfate content was higher in oasC under 15, but was lower in oasC under 750? I think the detailed explanation should be given, or discussed in the “Discussion” section.
Author Response
In this study, authors studied the function of OASC gene involved in sulfate uptake and translocation in Arabidopsis using the mutant. Additionally, they found a SNP in OASC gene among different accessions which is linked to the variation of sulfate content.
- In the “Materials and methods” section, the detailed primer sequences for the gene amplification and gene expression analysis should be provided. The Supplementary Table S1 did not provide the primers.
- Sorry for the omission, we added Supplementary Table S1 with the primers and renamed the other supplementary tables in the manuscript.
- The T-DNA insertion site of the mutants should be showed, as well as the gene expression level in mutants and wt plants.
- The insertion is shown as new Supplemental Figure 1, the expression, or rather the lack thereof, is shown in Fig. 3
- In Fig.5, why the sulfate content was higher in oasC under 15, but was lower in oasC under 750? I think the detailed explanation should be given, or discussed in the “Discussion” section.
- We have not been able to identify the mechanism of this observation, but we added a discussion on a link to sulphur use efficiency.
Reviewer 2 Report
The manuscript demonstrates the significance of OASC genes in the regulation of sulfate levels in Arabidopsis. This is a useful study in maintaining the sulfate levels in plants given the importance of sulfur in plant development and stress acclimation in the changing environmental conditions. The importance of the study is well highlighted with clear objectives in Introduction. However, the importance of sulfur may even be more elaborated in Introduction and Discussion. The study is suitable for inclusion in PLANTS. There are a few concerns that need to be addressed before formal acceptance of the manuscript.
How Mg from MgSO4 was maintained in the control and treated plants?
The study includes experiments done in controlled conditions and greenhouse where plants were grown in the soil. Include a statement to show the data obtained are comparable.
Please elaborate methodology for Cysteine and Glutathione determination.
Results are presented well and clearly written, but some date concerning sulfate are given in the Supplementary file. Please consider including Figure S1 in the main text.
Authors are advised to consider recently published papers in PLANTS and may consider to elaborate the contribution of sulfur in the regulation of mechanisms to maintain plant development and stress acclimation in Intro and Discussion. This will also enable update on the literature cited.
Authors concluded that loss of OASC gene for mitochondrial isoforms of OAS-TL results in a significant decrease in sulfate content. The approach is suitable in regulating sulfur levels in plants to withstand in the changing environmental conditions.
Author Response
The manuscript demonstrates the significance of OASC genes in the regulation of sulfate levels in Arabidopsis. This is a useful study in maintaining the sulfate levels in plants given the importance of sulfur in plant development and stress acclimation in the changing environmental conditions. The importance of the study is well highlighted with clear objectives in Introduction. However, the importance of sulfur may even be more elaborated in Introduction and Discussion. The study is suitable for inclusion in PLANTS. There are a few concerns that need to be addressed before formal acceptance of the manuscript.
- Thank you for the positive assessment. We added few words on importance of sulphur for crops in the discussion.
How Mg from MgSO4 was maintained in the control and treated plants?
- The explanation has been added to the methods.
The study includes experiments done in controlled conditions and greenhouse where plants were grown in the soil. Include a statement to show the data obtained are comparable.
- Actually, the conservation of the phenotype in both growth conditions is itself the evidence for its robustness. We added a sentence to the discussion.
Please elaborate methodology for Cysteine and Glutathione determination.
- We added more details to all analytical methods
Results are presented well and clearly written, but some date concerning sulfate are given in the Supplementary file. Please consider including Figure S1 in the main text.
- The root data were added to Figure 1
Authors are advised to consider recently published papers in PLANTS and may consider to elaborate the contribution of sulfur in the regulation of mechanisms to maintain plant development and stress acclimation in Intro and Discussion. This will also enable update on the literature cited.
- We added a citation from Plants in the discussion.
Authors concluded that loss of OASC gene for mitochondrial isoforms of OAS-TL results in a significant decrease in sulfate content. The approach is suitable in regulating sulfur levels in plants to withstand in the changing environmental conditions.
- We have added a short discussion on sulphur use efficiency.